# A neural basis for the spatial suppression of visual motion perception

Liu D Liu[1]*, Ralf M Haefner[2]*, Christopher C Pack[1]*

[1]Department of Neurology and Neurosurgery, Montreal Neurological Institute, McGill University, Montreal, Canada; [2]Department of Brain and Cognitive Sciences, University of Rochester, Rochester, United States

**Abstract** In theory, sensory perception should be more accurate when more neurons contribute to the representation of a stimulus. However, psychophysical experiments that use larger stimuli to activate larger pools of neurons sometimes report impoverished perceptual performance. To determine the neural mechanisms underlying these paradoxical findings, we trained monkeys to discriminate the direction of motion of visual stimuli that varied in size across trials, while simultaneously recording from populations of motion-sensitive neurons in cortical area MT. We used the resulting data to constrain a computational model that explained the behavioral data as an interaction of three main mechanisms: noise correlations, which prevented stimulus information from growing with stimulus size; neural surround suppression, which decreased sensitivity for large stimuli; and a read-out strategy that emphasized neurons with receptive fields near the stimulus center. These results suggest that paradoxical percepts reflect tradeoffs between sensitivity and noise in neuronal populations.

*For correspondence: liu.liu2@mail.mcgill.ca (LDL); ralf.haefner@gmail.com (RMH); christopher.pack@mcgill.ca (CCP)

**Competing interests:** The authors declare that no competing interests exist.

## Introduction

Perception relies on the spiking responses of sensory neurons. Indeed, individual neurons can exhibit exquisite selectivity for specific stimulus features. However, this single-neuron selectivity is of limited utility for stimulus encoding for two reasons. One is that neuronal responses are modulated by multiple stimulus dimensions, so that identical responses can be associated with very different stimuli. Another reason is that single-neuron responses can be quite variable, so that the response to the same stimulus can differ from one presentation to the next.

Some of this variability can be reduced by combining the responses of multiple neurons. If the variability is independent across neurons, it can be eliminated by simply averaging the responses of many neurons. In this case, the available information about the stimulus theoretically increases with neuronal population size (*Zohary et al., 1994*; *Shadlen et al., 1996*). However, in reality neuronal noise is usually correlated across nearby neurons, and such *noise correlations* are thought to greatly influence on the fidelity of a population code (*Abbott and Dayan, 1999*; *Sompolinsky et al., 2001*; *Panzeri et al., 1999*; *Averbeck et al., 2006*; *Ecker et al., 2011*). Still, current theories predict the stimulus information will increase or saturate as the size of the corresponding neuronal pool increases.

One simple way to manipulate the neuronal pool size is to change the physical size of a visual stimulus. Because neurons in early visual structures have small receptive fields, large stimuli recruit more neurons, potentially leading to more effective coding of stimulus properties and correspondingly better behavioral performance. It is therefore surprising that behavioral studies in humans have sometimes found that larger stimuli are associated with diminished perceptual performance (*Tadin et al., 2003*). Moreover, this *psychophysical suppression* of behavioral performance in human subjects is strongly correlated with various markers of mental function, including schizophrenia,

**eLife digest** People usually find it easier to see things when they are big and bright, but there are occasionally exceptions. One example concerns moving objects: when they are small, we can identify their direction of motion easily, but this becomes much more difficult for larger objects. This decreased perceptual sensitivity appears to be linked to other mental processes. For example, studies have suggested that people with high IQs have more difficulty perceiving the motion of large objects, whereas people with various psychiatric disorders, such as schizophrenia, are better able to see such movement. Although several theories have been proposed, there is currently no good explanation for these findings.

Liu et al. set out to determine why the part of the brain that is responsible for vision (the visual cortex) fails to register the direction of large moving objects and how this failure might relate to mental function in general. To do this, Liu et al. trained monkeys to report which direction different sized stimuli were moving on a screen. The electrical activity of nerve cells in the part of the visual cortex that deals with movement was recorded while the monkeys performed this task. The results of the experiments revealed that, on average, these cells responded strongly to large moving stimuli, even though the monkeys had trouble seeing their motion. However, nerve cells are "noisy" – they respond a bit differently every time they are presented with the same stimulus – and this noise was stronger for larger stimuli.

By studying the mathematical relationship between the noise and what the animals perceived, Liu et al. found that the visual cortex attempts to suppress the noise and in the process often shuts off the responses to large stimuli entirely. This suppression is likely to cause the movement of large stimuli to be poorly perceived.

If suppressing this kind of noise is really responsible for failures in perceiving motion, then this mechanism could also explain the connection between motion perception and other mental processes. Liu et al. are currently testing this idea.

major depression, and even I.Q (*Tadin et al., 2006*; *Golomb et al., 2009*; *Melnick et al., 2013*). These results have previously been hypothesized to reflect the strength of neuronal surround suppression in individual cortical neurons (*Tadin et al., 2003*; *Churan et al., 2008*), but it is unclear how such suppression affects neuronal populations, particularly in the presence of noise correlations.

To address this issue, we recorded from small populations of neurons in visual cortical area MT, in macaque monkeys trained to report the perceived direction of a moving stimulus. We varied stimulus size randomly from trial to trial, and found, as reported in human studies (*Tadin et al., 2003*), that increased stimulus size led to a drastic deterioration of behavioral performance. Our neurophysiological recordings revealed that the magnitude of the neuronal surround suppression of *individual* neurons is too small to account for psychophysical suppression. However, analysis of multi-electrode recordings revealed a novel aspect of neuronal noise correlations that further suppressed population coding for large stimuli: those neurons with the smallest surround suppression, and hence the ones most sensitive to large stimuli individually, also had noise correlations most closely aligned with signal correlations; such correlations are damaging to the total information carried by the population (*Abbott and Dayan, 1999*; *Averbeck et al., 2006*). These mechanisms, combined with conservative assumptions about the animals' behavioral strategies (*Pelli, 1985*; *Burr et al., 2009*; *Beck et al., 2012*), provided a full account of the observed psychophysical suppression. These results further our understanding of the relationship between neural activity and perception, in normal and pathological states.

## Results

In the standard model of perceptual decision-making (*Gold and Shadlen, 2007*), the responses of a population of sensory neurons are assumed to be read out by a decision-making area. For a linear read-out, this system is well-understood, and the key drivers of psychophysical performance are the sensitivities of the individual sensory neurons to the task-relevant stimulus dimension, their response variability, and the correlation structure in the population (*Zohary et al., 1994*; *Abbott and Dayan,*

*1999*; *Sompolinsky et al., 2001*; *Ecker et al., 2011*; *Moreno-Bote et al., 2014*). Since these quantities generally depend on the particular stimulus used for the task (*Snyder et al., 2014*) and the demands of the task itself (*Cohen and Newsome, 2008*), we performed simultaneous recordings from populations of MT neurons while two monkeys performed a task for which psychophysical surround suppression has previously been demonstrated in humans (*Tadin et al., 2003*).

In the remainder of this section we will first describe the psychophysical results, followed by our neurophysiological measurements. We then use the neurophysiological data to constrain a comprehensive model that can account for the observed pattern pf psychophysical suppression.

## Psychophysical measurements

We examined neuronal responses and behavioral performance during a task in which the visual stimulus size was varied across trials (*Figure 1A,C*) (*Tadin et al., 2003*). During the task, monkeys viewed drifting Gabor grating stimuli and reported their percepts of visual motion direction (*Britten et al., 1992*; *1996*) (*Figure 1C*). As in most human studies, we used a very brief stimulus duration (50 ms) (*Tadin et al., 2003*) in order to increase the difficulty of the task. In preliminary behavioral experiments we also compared psychophysical performance using Gabor patches of low (4%) and high (100%) contrast. Based on the dependency of the density of receptive fields on eccentricity in early visual structures (*Van Essen et al., 1981*; *Erickson et al., 1989*), we calculated that the number of visual cortical neurons activated by our stimulus should increase with stimulus size (*Figure 1B*).

Consistent with previous findings in humans (*Tadin et al., 2003*), we found that increasing the size of a low-contrast stimulus improved behavioral performance (*Figure 1D*, dashed lines), while under high-contrast conditions, behavioral performance worsened at larger sizes (*Figure 1D*, solid lines). Thus, paradoxically, psychophysical performance was best for stimuli of medium intensity, with performance declining as contrast and size were increased (*Figure 1D*, Wilcoxon rank sum test, p<0.001).

To quantify this effect, we computed a psychophysical suppression index ($SI_{psy}$) (*Figure 1D* and Material and methods), which captures the decrease in performance (on a scale from 0 to 1, with 0 corresponding to no suppression, and 1 to complete suppression) for large stimuli relative to the best performance across all stimuli. At 100% contrast, the $SI_{psy}$ of the psychometric function (mean ± s.d.) was 0.42 ± 0.25 for monkey C and 0.54 ± 0.19 for monkey Y, indicating that monkeys were approximately half as likely to accurately perceive the motion of a large stimulus, compared to a small one.

## Neurophysiological measurements

We recorded from small populations of neurons in MT using linear electrode arrays, while monkeys performed the high-contrast motion discrimination task described above. Area MT is thought to be causally involved in behavioral decisions for motion direction (*Born and Bradley, 2005*), and it contains many neurons with responses that are modulated by stimulus size and contrast (*Allman et al., 1985*; *Pack et al., 2005*). To maximize the number of stimulus repetitions per recording session, we fixed the stimulus contrast at 100% and varied stimulus size across trials. We analyzed data from 165 single units, with 2–8 cells being available on any given day.

## Relationship between single-neuron selectivity and behavior

The responses of an example MT neuron to stimuli centered on the receptive field are shown in *Figure 2A*. Here the orange and violet dots show the responses to the preferred and null direction stimuli, and these responses decrease slightly with increasing stimulus size. The distributions of these responses across trials can be converted into a single measure of neuronal selectivity, d′, which is plotted as a function of stimulus size in *Figure 2B*. Based on this *neurometric function*, we can compute a neural measure of suppression, $SI_{neu}$, which is defined analogously to $SI_{psy}$. The value of $SI_{neu}$ for this neuron was 0.18, which indicates a modest suppression of motion signaling by large stimuli.

The decrease in neuronal selectivity with stimulus size resembles the psychophysical performance of the monkey (*Figure 2B*). However, the strength of neuronal surround suppression is substantially less than that of the simultaneously measured psychophysical suppression (0.54). This was often the case in our data: For the MT population, the mean neuronal d′ ($SI_{neu}$ = 0.27) was much less suppressed than the mean psychometric d′ ($SI_{psy}$ = 0.48, *Figure 2D*). Moreover, many neurons exhibited

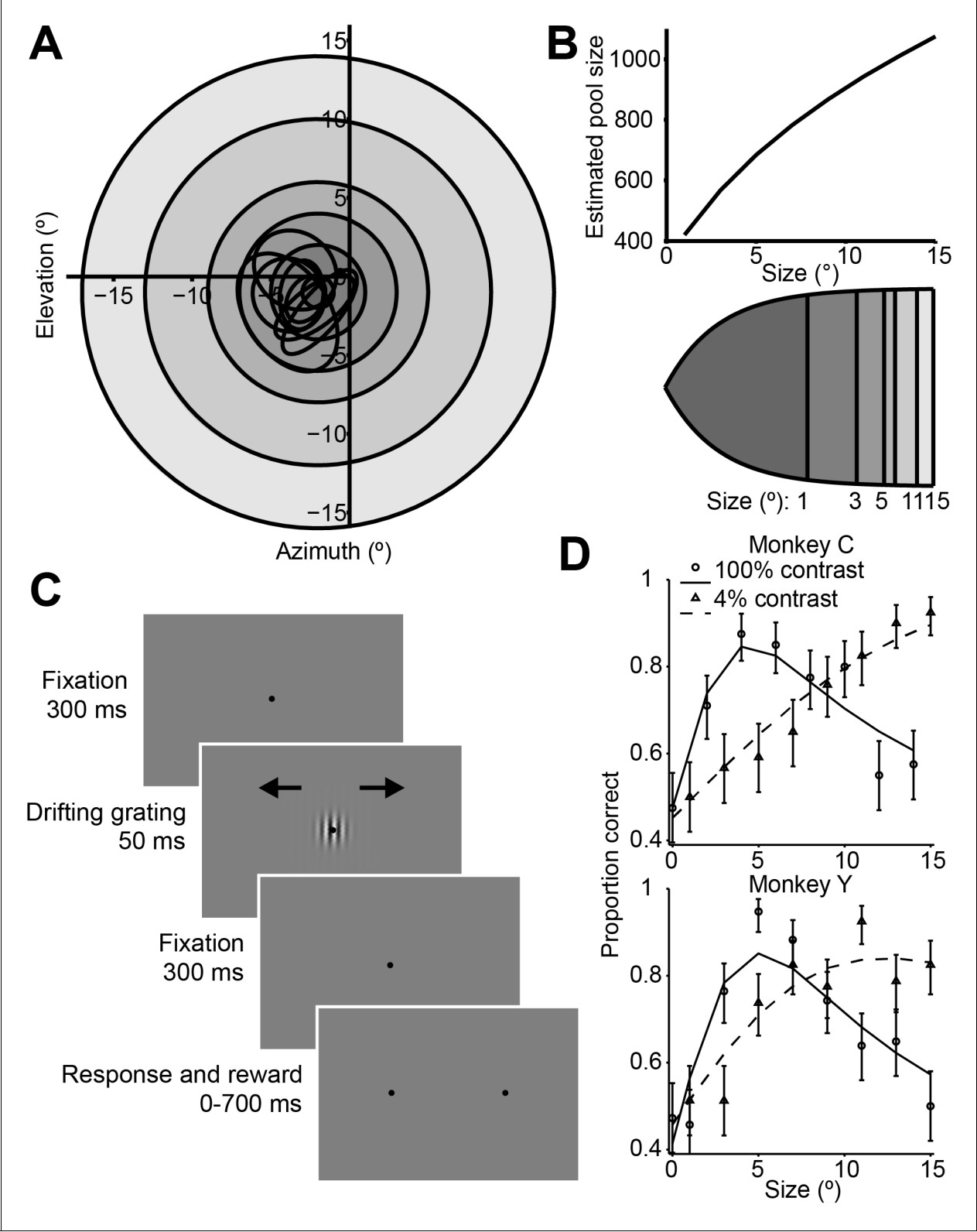

**Figure 1.** Stimuli, sequence of events, and behavioral performance in the task. (**A**) Receptive fields from an example recording session, shown as black ovals, relative to lines of different visual eccentricity (gray circles) commensurate with the stimulus sizes used in the experiments. (**B**) The estimated neuron pool size as a function of stimulus size, for the eccentricities and stimulus sizes used in the experiments (top). Cortical mapping of visual space from (**A**), showing that larger stimuli projected onto larger extents of cortical space (bottom). The sizes of the shaded areas correspond to the

*Figure 1 continued on next page*

*Figure 1 continued*

estimated cortical footprint (see Materials and methods). (C) Behavioral task. The animals were required to maintain fixation in a 2° window for 300 ms, after which a drifting Gabor appeared briefly. Animals were then required to fixate for another 300 ms until the fixation spot disappeared. The animals then indicated their motion percept with an eye movement to one of two targets within 700 ms. (D) Examples of the animals' psychometric functions for high contrast (solid line, circles) and low contrast (dashed line, triangles) stimuli. Error bars represent 95% binomial proportion confidence interval.

no surround suppression at all, even for stimuli extending beyond their receptive fields (*Born and Tootell, 1992*), and the selectivity of these neurons to large stimuli routinely exceeded that of the monkeys (*Figure 2B,C*). This was especially clear in neurons with receptive fields near the edges of the larger stimuli (*Figure 2—figure supplement 1B*); in these neurons responses increased with stimulus size (*Figure 2—figure supplement 1C*). Together these results suggest that the psychophysical performance is not solely driven by typical single-neuron selectivity, as only a small fraction of neurons showed suppression comparable to that of the behavior.

One caveat to this conclusion is that subjects might have relied more heavily on a subpopulation of MT neurons to form their perceptual decisions. Indeed, if neurons with strong surround suppression exerted a greater influence on perception, perhaps by virtue of anatomical connectivity (*Born et al., 2000*; *Berezovskii and Born, 2000*), then psychophysical suppression would presumably increase accordingly. However, using choice probability analysis (*Britten et al., 1996*; *Nienborg et al., 2012*; *Haefner et al., 2013*), we found no evidence that neurons with strong surround suppression were more correlated with the animals' behavior choices; indeed the correlation between $SI_{neu}$ and choice probability was modestly negative (*Figure 2E*; $r = -0.14$, $P = 0.04$).

## Noise correlation measurements

The mean levels of noise correlations were typically on the order of 0.1 ($0.099 \pm 0.007$), compatible with previous reports (*Zohary et al., 1994*; *Bair et al., 2001*; *Huang and Lisberger, 2009*). Their strength was independent of motion direction or stimulus size (Wilcoxon rank sum test for direction, 94% of experiments with p>0.05; for smaller and larger sizes, $P = 0.55$ *Figure 4—figure supplement 1A*).

Next, we considered the relationship between noise correlations and tuning curve similarity; these have been found to correlate in previous studies (*Bair et al., 2001*; *Huang and Lisberger, 2009*). *Figure 3A* shows the responses of two example neurons that were recorded simultaneously; each dot represents the mean response to a preferred (red) or null (blue) direction stimulus, with different dots corresponding to responses to different stimulus sizes. The responses of these neurons exhibit a clear signal correlation ($r_{signal} = 0.61$). *Figure 3B* shows trial-by-trial data from the same pair of neurons; here the responses have been z-scored to remove changes in the mean due to different stimulus sizes or directions (*Zohary et al., 1994*). The remaining dependency reflects noise correlations in the responses of the two neurons ($r_{noise} = 0.21$). The relationship illustrated by this example pair is characteristic of the population (*Figure 4A*), across which noise correlations and signal correlation are significantly correlated ($r = 0.32$, p<0.001).

Interestingly, we find that this *correlation structure* appears to be different for pairs of neurons with different levels of surround suppression. This is apparent in the examples shown in *Figure 3*. To study this relationship across the population (N = 370 pairs), we classified neurons as surround suppressed (SS) or not (NS), based on a simple median split of the $SI_{neu}$ distribution (*Glasser et al., 2011*) (*Figure 2C*). This yielded three types of neuron pairs: both suppressed (SS-SS), both non-suppressed (NS-NS), and mixed (SS-NS). Across the population, the magnitudes of $r_{noise}$ were not significantly different across types of neuron pairs (Wilcoxon rank sum tests, p>0.86). However, the correlation structure differed substantially for different cell classes: For the NS-NS pairs, $r_{noise}$ and $r_{signal}$ tended to be correlated (*Figure 4A*, red dots). By contrast pairs of SS neurons showed less of a dependency of noise correlation on signal correlation (*Figure 4A*, blue dots). The difference in the slopes of the lines relating signal and noise correlations was significantly lower for the SS pairs than for the NS pairs (ANCOVA, $P = 0.03$, multiple comparison test) (*Figure 4A*, red and blue lines). For NS-SS pairs, this dependency was intermediate (*Figure 4A*, black line).

We performed several control analyses to verify that these results reflected a genuine difference in correlation structure across cell types. First, we recalculated $r_{signal}$ using direction tuning curves

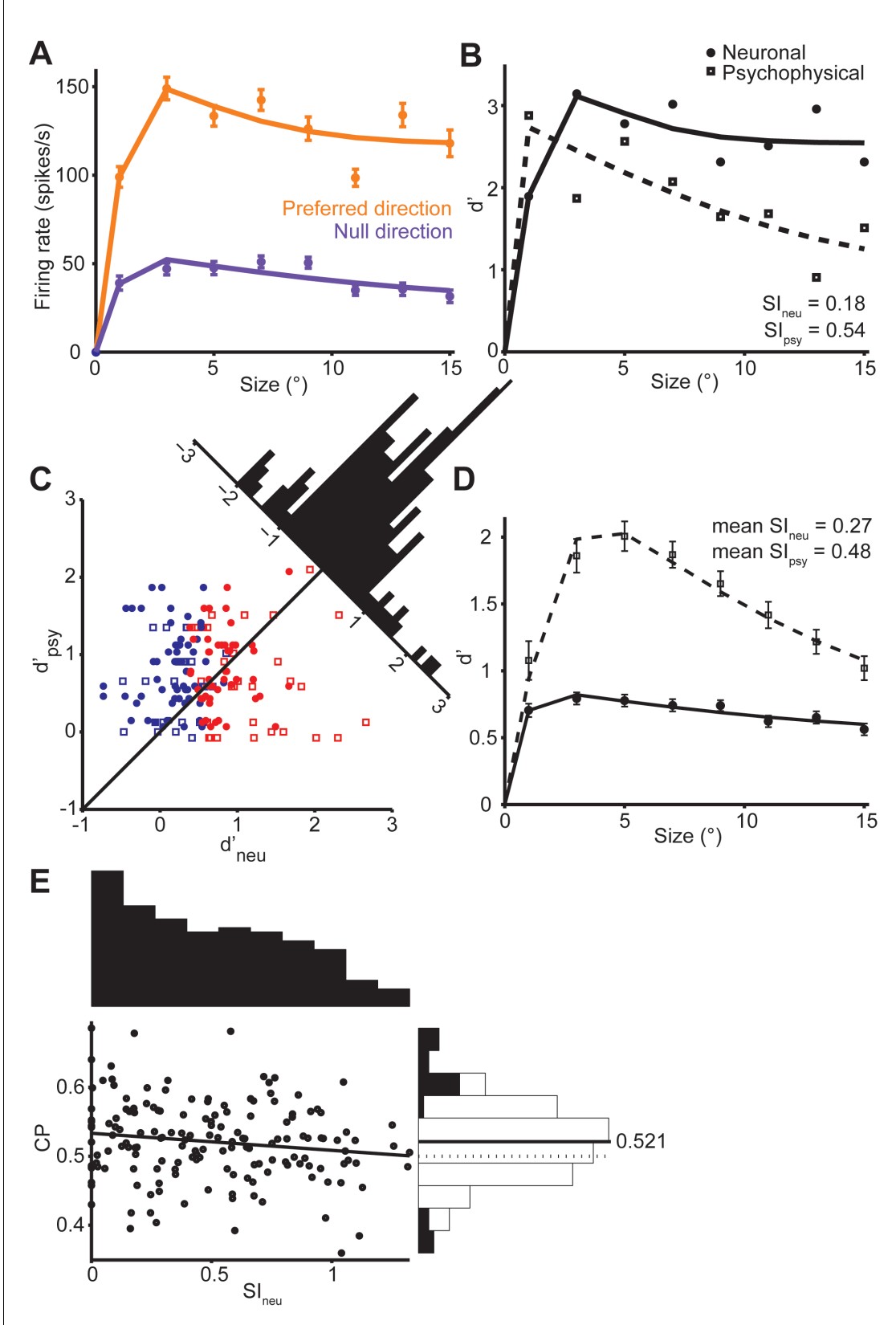

**Figure 2.** Quantification of single neuron selectivity for an example MT neuron, and the summary for the population. (**A**) Size tuning curves, plotting the firing rate (mean ± s.e.m.) for the preferred (orange) and null (violet) direction stimuli as a function of Gabor patch size. The lines indicate difference of

*Figure 2 continued on next page*

*Figure 2 continued*

error functions fits. (B) Neurometric function (filled symbols) for the example neuron plotting the d' value as a function of stimulus size. The corresponding psychometric function is superimposed (open symbols). Solid and dashed lines indicate difference of error functions fits. The psychophysical performance differs from *Figure 1D* since the stimulus was tailored to the neural population measured on any one day. (C) Scatter plot of the psychophysical d' against the neuronal d' at the largest stimulus size. Filled circles represent monkey C ($n = 105$), and open squares represent monkey Y ($n = 60$). Red represents neurons with weak surround suppression, and blue represents neurons with strong surround suppression. The distribution of $d'_{neu}-d'_{psy}$ is shown at the diagonal. (D) The mean $d'_{psy}$ as a function of size from all sessions (monkey C: $n = 28$, monkey Y: $n = 11$) superimposed with the mean single neuron $d'_{neu}$ from all MT neurons (165 single neurons). Error bars denote s.e.m. (E) Population summary of choice probability (CP). Scatter plot of CP against the suppression index of the neurometric function. Filled symbols represent CP values that are significantly different from 0.5 (p<0.05, permutation test). Solid line indicates linear fit ($r = -0.14$, $P = 0.04$). The marginal distributions of $SI_{neu}$ and CP are shown on the top and the right. Filled and open bars indicate neurons with significant and non-significant choice probabilities, respectively.

The following figure supplements are available for figure 2:

**Figure supplement 1.** (A) RF positions of the neurons recorded.

**Figure supplement 2.** Quantification of choice probability (CP) of single neurons and the time course of CP.

that were measured for a fixed stimulus size. This controlled for any variation in $r_{signal}$ that arose from differences in the size-tuning functions of NS and SS neurons. The results (*Figure 4—figure*

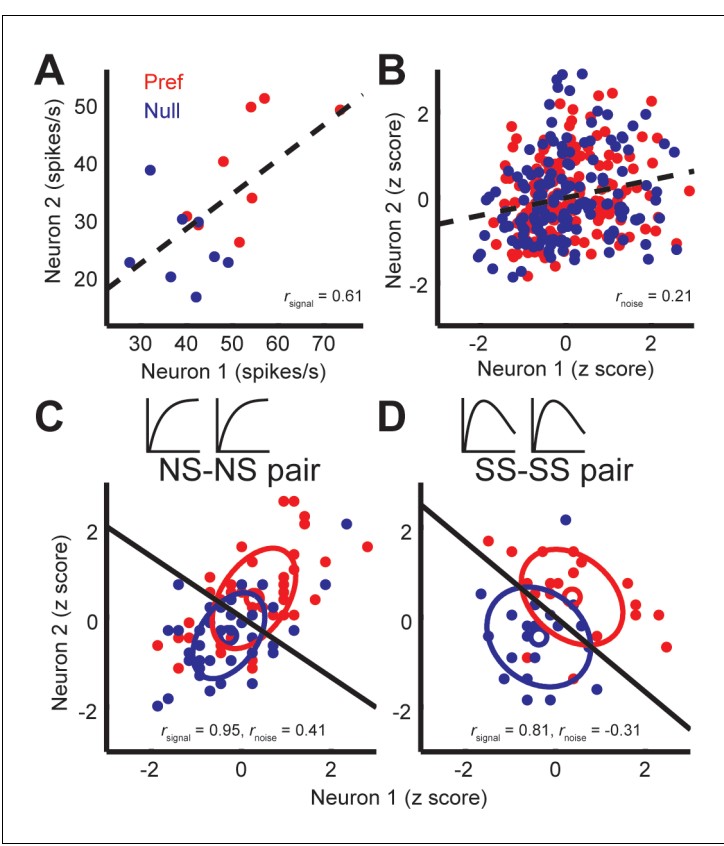

**Figure 3.** Quantification of noise correlation ($r_{noise}$) and signal correlation ($r_{signal}$) between neuron pairs. (A) The mean responses of the two simultaneously recorded neurons across both directions and sizes. $r_{signal}$ (0.61) is the Pearson correlation coefficient of the mean responses for the conditions. (B) The responses for each stimulus condition were z scored across the repetitions, and each point represents a response from one trial. $r_{noise}$ (0.21) is the Pearson correlation coefficient of the entire dataset. The dashed lines represent linear fits. (C, D) Response correlations for an NS-NS pair and an SS-SS pair for one example stimulus size (1°).

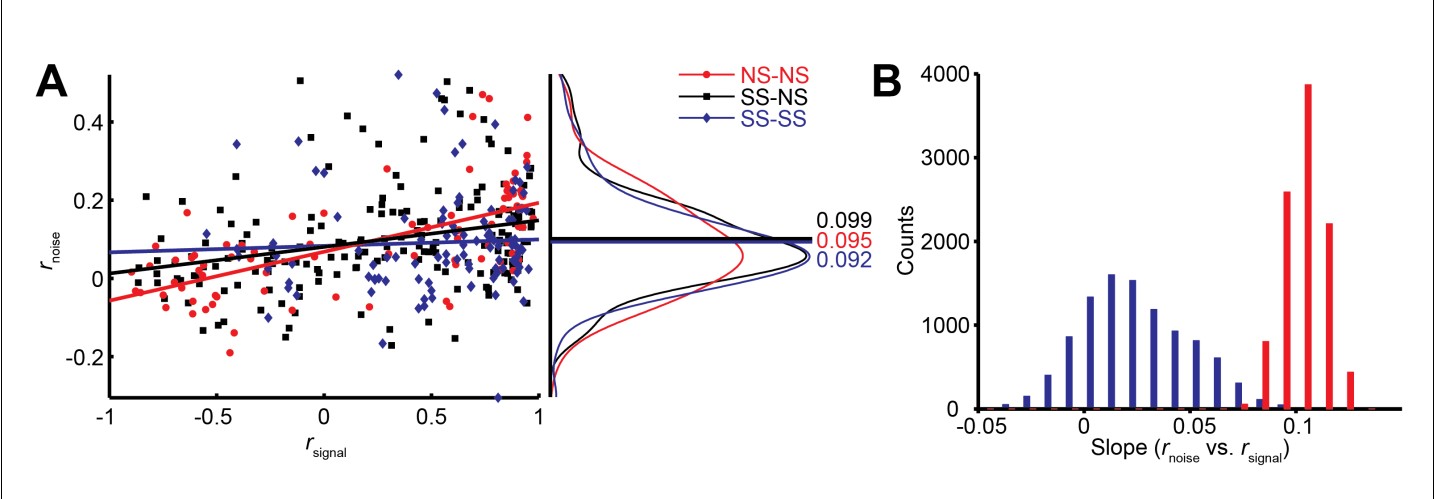

**Figure 4.** Relationship between noise correlation ($r_{noise}$) and signal correlation ($r_{signal}$). (A) Scatter plot of $r_{noise}$ versus $r_{signal}$ for pairs of SS and SS (blue), SS and NS (black), and NS and NS (red) neurons. Lines represent linear regression fits. Marginal distributions of $r_{noise}$ are also shown (right panel). Lines and numbers mark the mean values of $r_{noise}$ for each combination of neuron pairs. (B) Sampling from rate matched sub-distributions of SS-SS and NS-NS pairs gives similar differences in $r_{noise}$ vs. $r_{signal}$ slope.
The following figure supplement is available for figure 4:

**Figure supplement 1.** Effects of stimulus conditions, firing rate, and tuning similarity on the $r_{noise}$ on $r_{signal}$ dependency.

*supplement 1D*) were similar to those in *Figure 4A* (ANCOVA, $P = 0.04$, multiple comparison test). Second, we verified that these results were not due to changes in firing rate across the different cell types, as the mean firing rates of NS-NS pairs (median = 39.1) and SS-SS pairs (median = 36.4) were not significantly different (Wilcoxon rank sum test, $P = 0.45$). Also, sampling from rate-matched sub-distributions of SS-SS and NS-NS pairs (Materials and methods) yielded significantly higher $r_{noise}$ vs. $r_{signal}$ slopes for the NS-NS sub-distributions (*Figure 4B*; Wilcoxon rank sum test, p<0.001). Finally, the reduction of this $r_{noise}$ dependency did not depend on the categorical classification of SS and NS neurons, as we obtained similar results using continuous values of the joint $SI_{neu}$ for pairs of neurons (*Figure 4—figure supplement 1C*; linear correlation: ($r = -0.232$, p<0.0001). This finding suggests that the correlated variability between two neurons with similar stimulus preferences may largely arise from the same inputs that are responsible for surround suppression in those neurons.

Differential correlations (*Moreno-Bote et al., 2014*) between neurons i and j are those that are proportional to $f_i'f_j'$, where $f_i$ denotes the tuning function of neuron i, and the prime denotes the derivative with respect to the task-relevant direction in stimulus space; such correlations will limit the information carried even for arbitrarily large neural populations (*Moreno-Bote et al., 2014*). We calculated differential correlations for all neuronal pairs, and found that there is indeed a positive relationship between noise correlations and $f'f'$ (*Figure 4—figure supplement 1E*). Furthermore, we find the same difference between SS-SS and NS-NS pairs as reported above (*Figure 4A*): the magnitude of the information-limiting correlations is greater between NS-NS pairs than between SS-SS pairs (*Figure 4—figure supplement 1E*, $r_{NS-NS} = 0.48$, $r_{SS-SS} = 0.23$, $P = 0.08$). In brief, while NS neurons are individually more informative for large stimuli than SS neurons, as a population they are more limited by their correlation structure than SS neurons.

## Modeling results

Based on our empirical measurements described above, we devised a model to investigate to which degree each aspect of the neural data contributed to the observed psychophysical behavior. Such modeling is naturally limited by the impossibility of measuring the relevant properties of all the sensory neurons involved in processing the stimuli. Thus we accounted for this uncertainty explicitly by examining a large number of models from a joint probability distribution over parameters

corresponding to the properties of the MT population response (e.g, firing rates, noise correlations, direction tuning bandwidth, etc.).

A detailed description of the modeling approach is given in the methods. In brief, we generated populations of synthetic neurons by sampling neural properties from a joint truncated Normal distribution over tuning curve parameters inferred from our measurements. In that way we could simulate neural populations that not only matched the observed marginal statistics but also the correlations between the measured parameters (*Figure 5—figure supplement 1*). Since we only measured neurons with RFs covering approximately the central 5° of the stimulus, we extrapolated from these neurons to those at larger eccentricities by shifting the size tuning curves of our measured neurons according to the distance between the simulated RF and the center of the stimulus. Furthermore, we scaled the number of model neurons according to the observed dependency of the magnification factor on eccentricity (*Van Essen et al., 1981*; *Erickson et al., 1989*). We sampled the noise correlation structure from a Wishart distribution around the empirical means as a function of the signal correlation between neuron pairs (*Figure 4A*). By generating many such populations for each model, we extracted a range of predictions of behavioral performance for different stimulus sizes (represented by the error bars in *Figure 5—figure supplement 3A*), so that for each model we could compute its range of predicted psychophysical suppression (*Figure 5—figure supplement 3B*). The predicted model suppression is the key metric that we are interested in, and its dependency on the key model parameters are explored in *Figure 5— figure supplement 4* and *Figure 5D*.

In order to relate our simulated neural responses to behavioral performance (*Figure 5A*) we used a standard linear read-out in which a weighted average of the responses is compared to a decision-threshold (*Shadlen et al., 1996*; *Gold and Shadlen, 2007*; *Haefner et al., 2013*; *Smolyanskaya et al., 2015*; *Pitkow et al., 2015*). We made the assumption of a factorial decoder (*Figure 5B*), in which the read-out weight for each neuron only depends on the properties of that neuron itself, for two primary reasons: First, such a set of read-out weights can be learned easily since each weight only depends on the properties of the individual neuron itself (*Law and Gold, 2009*), and second, it has recently received empirical support (*Pitkow et al., 2015*). (We also performed our analysis using an optimal linear read-out, as well as one in which each neuron's weight depended only on its sensitivity to the stimulus and not its variability, and obtained qualitatively similar results – see Supplementary Information, *Figure 5—figure supplement 2*). Since the stimulus size in our experiment is randomized, and since the duration is extremely brief (50 ms), we furthermore assume that the read-out is fixed and does not adjust dynamically to the stimulus size. We initially limited the read-out to neurons with receptive fields within 5° of the stimulus center; we examine the impact of this choice below.

*Figure 5C* shows the average performance over 100 runs of this model. As in the behavioral data, we find that performance decreases for larger stimuli. The suppression shown by the model is of the same magnitude as the empirical behaviour (*Figure 5E*, black), with the model SI being 0.48 (*Figure 5C,E* cyan). To understand the source of this suppression, we performed additional analyses in which key components of the model were removed: Specifically, we considered models in which (*Zohary et al., 1994*) noise correlations were absent; (*Shadlen et al., 1996*) correlations were as measured, but surround suppression was absent; and (*Abbott and Dayan, 1999*) correlations and surround suppression were on average as measured, but the observed relationship between them (*Figure 4A*) was missing. We found that both the noise correlation structure and surround suppression were necessary to account for the decreased performance as a function of size, since models (*Zohary et al., 1994*) and (*Shadlen et al., 1996*) did not show any psychophysical suppression at all (SI = 0; data not shown). However, these components together were not sufficient to account for the observed behavioral results, since model (*Abbott and Dayan, 1999*) exhibited only modest psychophysical suppression (SI = 0.28; *Figure 5C,E* magenta). Thus the relationship between surround suppression and correlation structure appears to have important consequences for motion perception.

From *Figure 5C* (cyan) it is apparent that the surround-suppression-dependent correlation structure has two separate effects on performance: One is a suppression of motion signal for large sizes. Perhaps more surprising is an increase in performance seen for small stimuli (*Figure 5C,E*). This suggests that the combined effect of correlation structure and surround suppression is an increase in the capacity of the MT population to discriminate the direction of very small stimuli, at the expense of large stimuli (see Discussion).

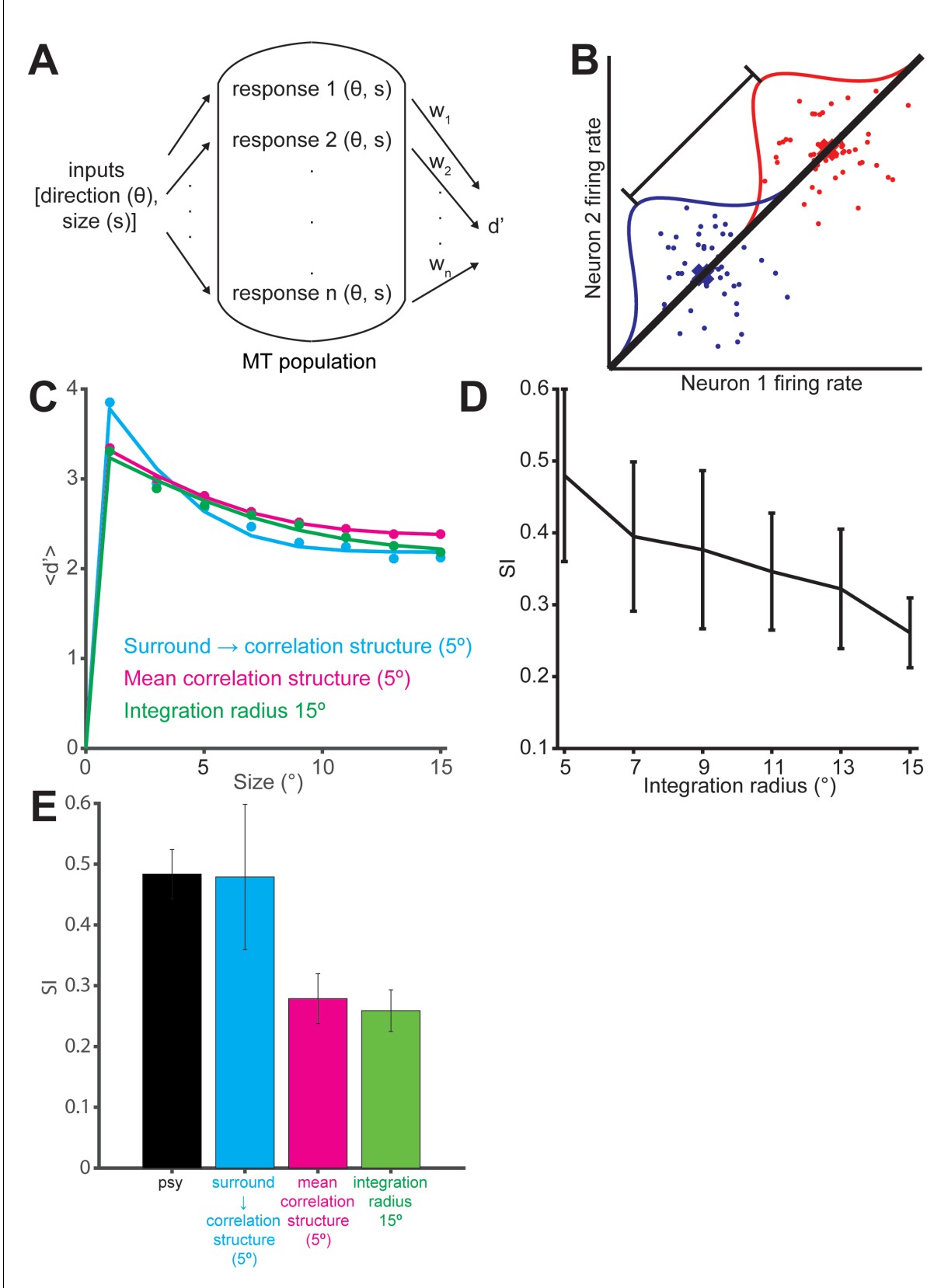

**Figure 5.** Simulation of population selectivity and model comparisons. (**A**) Schematic of the population selectivity simulation. The preferred and null responses were sampled from the distribution of parameters recorded. We tested combinations of different correlation structures and readout weights.

*Figure 5 continued on next page*

*Figure 5 continued*

(B) Calculation of the population selectivity. Each point represents the response from a trial from *n* neurons (here, *n* = 2). The one-dimensional distributions for the preferred and null direction responses were generated by projecting the points onto the normalized axis that connects the mean responses in *n*-dimensional space (factorial read-out). The calculation of population d' then follows the equation in the Materials and methods. (C) The predicted population neuronal selectivity plotted as a function of the stimulus size for each model. Data points represent averages across 100 iterations of the simulation, with each iteration based on a different re-sampling of the parameter set from the original data sets. Results are shown for the full model based on all empirical measurements in which surround suppression modulates noise correlation structure (cyan); a model where correlations and surround suppression were on average as measured, but the observed relationship between them was missing (magenta), and finally the full model again, but with integration radius of 15° (green). (D) SI of population d' for simulations where increasing number of neurons with peripheral receptive fields are included. The x-axis indicates the integration window for including neurons' receptive fields relative to the stimulus center. Error bars denote standard deviation. (E) Comparison of the SI of different simulations: colors as in C. Error bar for the psychophysical data denotes s.e.m., and error bars for the model predictions denote standard deviation.

The following figure supplements are available for figure 5:

**Figure supplement 1.** Distributions of the parameters for the Difference of Error functions fits.

**Figure supplement 2.** Comparison of the SI of different decoders.

**Figure supplement 3.** Distributions of measured and simulated behavioral d' and SI.

**Figure supplement 4.** Predicted model SI as other model parameters were varied.

The preceding analyses suggests that psychophysical suppression is due to a combination of two known aspects of neural coding, surround suppression and noise correlations. Equally important is novel interaction between these two factors, wherein the correlation between neurons depends on their respective surround suppression (*Figure 4A*). To arrive at these conclusions, we assumed that the animals used a fixed read-out, focusing on neurons with receptive fields near the center of the stimuli. To determine the importance of this assumption, we ran model simulations in which the integration radius was varied (*Figure 5D*). Unsurprisingly the SI decreased with increasing integration radius, dropping to 0.26 when the radius was 15°, which is significantly less than that exhibited psychophysically by the monkeys (*Figure 5E* green, Wilcoxon rank sum test, p<0.001). The overall model performance, obtained by summing the performance across all sizes, was, however, unaffected by this parameter (ANOVA, *P* = 0.25). This is due to the fact that a larger integration radius increases the performance at large sizes, while decreasing the performance at small sizes (*Figure 5C*). This suggests that behavioral SI could vary substantially according to the internal strategies used by the observer.

## Discussion

Using multi-electrode recordings in combination with a behavioral task, we have examined the effects of stimulus size on population coding. Consistent with previous work (*Zohary et al., 1994*; *Huang and Lisberger, 2009*; *Cohen and Newsome, 2009*), we find that pairs of MT neurons exhibit modest noise correlations, with typical correlation coefficients near 0.10. We also find that the strength of noise correlations is related to the strength of signal correlations and that this relationship limits the benefit of increasing stimulus size on population coding. Moreover, we find that the correlation structure is not constant across MT neuron pairs, but rather is related to the strength of a seemingly unrelated variable, surround suppression. This relationship between signal correlations, noise correlations, and stimulus selectivity appears to have two important effects on visual perception: Large stimuli are encoded poorly because of a strong decrease in selectivity for surround-suppressed neurons, and undesirable noise correlations in non-suppressed neurons. Meanwhile small stimuli are encoded more effectively because of the combination of strong direction selectivity and advantageous correlation structure in surround-suppressed MT neurons. Below we suggest that this population size tuning might have important implications for perception and behavior.

## Comparison to previous studies of noise correlations

Surround suppression has often been hypothesized to reduce correlations in natural inputs (*Snyder et al., 2014*; *Vinje and Gallant, 2002*). We find that neurons with strong surround suppression can exhibit larger or smaller noise correlations, depending on the strength of their signal correlations. This relationship holds for all stimuli, even those that do not engage the receptive field surrounds strongly.

Previous studies have shown that the magnitude of $r_{noise}$ is not fixed, but can be reduced by adaptation (*Gutnisky and Dragoi, 2008*), learning (*Gu et al., 2011*), and attention (*Cohen and Maunsell, 2009*; *Mitchell et al., 2009*). The latter is particularly relevant, because attention increases the effective contrast of the stimulus (*Treue and Trujillo, 1999*), which also increases surround suppression (*Sundberg et al., 2009*) and decreases correlations (*Kohn and Smith, 2005*). Thus a single mechanism (*Reynolds and Heeger, 2009*) may account for the effects of attention and surround suppression on noise correlations, as implemented with divisive normalization (*Tripp, 2012*; *Wiechert et al., 2010*). Attention is also of interest because, like surround suppression, it can increase or decrease the strength of noise correlations, depending on the stimulus encoding of the neuron pairs (*Ruff and Cohen, 2014*). These differential effects on positive and negative noise correlations are particularly important in MT, where negative correlations are quite common (*Zohary et al., 1994*; *Huang and Lisberger, 2009*). Negative correlations likely arise from motion-opponent mechanisms, in which the outputs of neurons with opposite direction tuning are subtracted. Such effects are stronger in MT than in V1 (*Qian and Andersen, 1994*), and they play an important role in decision-making models (*Shadlen et al., 1996*; *Cohen and Newsome, 2009*).

The results shown in *Figure 5D* suggest that incorporating the responses of a limited number of the MT neurons also contributed to psychophysical suppression. In a technical sense such a strategy is suboptimal (*Beck et al., 2012*), as subjects could probably have performed better by making use of the neurons with receptive fields near the edges of the stimulus. Although we have no direct measure of the actual readout strategy used by the subjects, we suggest that the limited sampling used here is a more realistic model of the neural decision process, for several reasons. First, recall (*Figure 1C*) that stimuli sizes were randomly interleaved, so that motion information was always present in central locations, but for peripheral locations it was only present for large stimuli. Previous work suggests that subjects allocate resources according to the uncertainty associated with individual stimulus positions (*Pelli, 1985*), so that monkeys in our task likely made greater use of neurons with receptive fields positioned near the center of the stimulus. In addition, although the subjects could have used neurons with receptive fields positioned near the edge of the stimulus to extract additional information about the motion of large stimuli (*Tsui et al., 2010*), we found instead that choice probability decreased with receptive field eccentricity (*Figure 2—figure supplement 2D*; $r = -0.48$, $P = 0.05$). This suggests that the monkeys likely based their decisions on neurons with receptive fields closer to the center of the stimulus, where motion information was present reliably on every trial. It would therefore be interesting to study psychophysical suppression in a paradigm in which the stimulus location was unpredictable from trial to trial. We predict that psychophysical suppression would be reduced in this case, as would overall performance across sizes (*Herrmann et al., 2010*).

A related possibility is that the subjects made use of a suboptimal decoding strategy (*Moreno-Bote et al., 2014*; *Pitkow et al., 2015*). Indeed our analyses were based on a standard factorial decoder (*Abbott and Dayan, 1999*; *Sompolinsky et al., 2001*; *Ecker et al., 2011*), which ignores correlation structure and hence loses information. We have reanalyzed our results using an optimal linear decoder (*Moreno-Bote et al., 2014*; *Pitkow et al., 2015*; *Salinas and Abbott, 1994*), and found that this approach does improve performance in general. However, the main conclusions with respect to correlation structure and its dependence on surround suppression are unchanged (*Figure 5—figure supplement 2*).

## Perceptual correlates of surround suppression

The paradoxical decline in motion perception with increasing stimulus size, first observed in human psychophysics (*Tadin et al., 2003*), has often been attributed to neuronal surround suppression at the level of MT. Indeed, transcranial magnetic stimulation (TMS) that targets MT reduces the spatial suppression effect (*Tadin et al., 2011*). However, the TMS protocols used to modulate spatial

suppression are inhibitory, and so one might just as easily interpret these results as an effect on noise correlations (*Waterston and Pack, 2010*). This interpretation is consistent with our results, assuming that inhibitory connectivity plays a role both in generating surround suppression and in regulating noise correlations (*Tripp, 2012*; *Carandini and Heeger, 2012*; *Renart et al., 2010*).

The distinction is important in interpreting a large body of data showing reduced spatial suppression in certain human populations. Examples include people with schizophrenia (*Tadin et al., 2006*), and older individuals (*Betts et al., 2005*). Although these subjects may have deficits in GABAergic efficacy (*Tadin et al., 2006*; *Betts et al., 2005*), our results suggest that the connection to psychophysical spatial suppression could also be through noise correlations, as these are necessary to produce any effect of neural surround suppression at the population level.

### Optimal encoding of small stimuli and pursuit targets

Our simulation results suggest that surround suppression can increase the selectivity of the neuronal population to the smallest stimulus size in this task, while worsening the selectivity at larger sizes (*Figure 5C*; note performance for the 1° stimulus). Therefore, one benefit of surround suppression might be in the tracking of small moving stimuli. Indeed, activity in clusters of surround-suppressed neurons has been found to be causally linked to the tracking of small targets in smooth pursuit (*Born et al., 2000*).

The link between MT activity and smooth pursuit initiation has been further strengthened by the finding that neuronal variability in MT can account for the majority of motor variation in smooth pursuit (*Osborne et al., 2005*; *Hohl et al., 2013*). These observations have led to the suggestion that correlation structure in MT might limit the precision of pursuit initiation (*Huang and Lisberger, 2009*). Our results suggest that such comparisons should take into account the center-surround properties of individual MT neurons, as the neurons that seem to contribute most directly to pursuit initiation (*Born et al., 2000*) exhibit more advantageous correlation structure (*Figure 4A*). As a result, the pursuit initiation system might benefit from averaging the activity of many surround-supressed MT neurons. This would explain both the weak correlation between single-neuron MT activity and pursuit and the relatively low choice probability of surround suppressed neurons in our perception task (*Figure 2E*).

It is interesting in this regard that some models of smooth pursuit initiation (*Hohl et al., 2013*) involve both a motion opponency step and a normalization operation. Normalization in these models serves the function of computing a vector average of the MT population response, and it also affects the levels of noise correlations in a manner that accounts for trial-to-trial fluctuations in behavior. Our results suggest the additional function of reshaping the selectivity of the MT population response in such a way as to favor the motion of small stimuli, precisely as would be expected for a system that initiates orienting responses to moving objects in a natural environment (*Lettvin et al., 1959*).

## Materials and methods

### Subjects and apparatus

Two adult female rhesus monkeys (*Macaca mulatta*, both 7 kg) were used for electrophysiological recordings in this study. Before training, under general anesthesia, an MRI-compatible titanium head post was attached to each monkey's skull. The head posts served to stabilize their heads during subsequent training and experimental sessions. For both monkeys, eye movements were monitored with an EyeLink1000 infrared eye tracking system (SR Research) with a sampling rate of 1000 Hz. Visual motion stimuli were displayed at 60 Hz at 1280 by 800 pixels resolution; the viewing area subtended 60° × 40° at a viewing distance of 50 cm. The sizes of the Gabor patches were defined by 2 standard deviations of the Gaussian envelope and ranged from 1° to 15° in steps of 2°. All procedures conformed to the regulations established by the Canadian Council on Animal Care and were approved by the Institutional Animal Care Committee of the Montreal Neurological Institute.

### Electrophysiological recordings

Area MT was identified based on an anatomical MRI scan, as well as depth, prevalence of direction-selective neurons, receptive field size to eccentricity relationship, and white matter to grey matter

transition from a dorsal-posterior approach. We recorded single-units using linear microelectrode arrays (V-Probe, Plexon) with 16 contacts. Neural signals were thresholded online, and spikes were assigned to single units by a template-matching algorithm (Plexon MAP System). Offline, spikes were manually sorted using a combination of automated template matching, visual inspection of waveform, clustering in the space defined by the principle components, and absolute refractory period (1 ms) violations (Plexon Offline Sorter).

## Stimulus and Discrimination task

Animals were trained to perform coarse motion direction discrimination tasks with Gabor patches. The structure of an individual trial is illustrated in *Figure 1C*. Each trial began with the onset of a fixation point. The monkey was required to establish and maintain fixation within a 2° × 2° window for 300 ms, after which a drifting Gabor patch appeared on the receptive field centers. The parameters of the Gabor patch were matched to the multi-unit preferences for spatial position, preferred direction, and spatiotemporal frequency (*Figure 1A* and *Figure 2—figure supplement 1A*). We included all units that exhibited significantly different responses (t-test; p<0.05) to their preferred and null directions at the smallest stimulus size, and a preferred direction within ± 42° of one of the directions of the stimulus used for behavioral testing. The range of stimulus sizes (0–15° radius at 2.3 ± 0.5° eccentricity) was chosen to straddle the receptive field sizes (2.2 ± 1.1° radius at 3.2 ± 1.3° eccentricity) of the recorded neurons (*Figure 1A* and *Figure 2—figure supplement 1A*).

The motion stimulus was presented for a brief period (typically 50 ms), after which the monkey was required to maintain fixation for another 300 ms. The fixation point then disappeared, and two choice targets appeared, after which the monkey made a saccade to the corresponding target to report its perceived motion direction (preferred or null relative to the neuron isolated). The monkey was required to indicate its decision within 700 ms following the onset of the choice targets. Correct choices were rewarded with a drop of liquid. If fixation was broken at any time during the stimulus, the trial was aborted. In a typical session, the monkeys performed 20–40 repetitions of each distinct stimulus.

## Data analysis

The psychophysical d′ was calculated as

$$d'_{psy} = z_{hit\ rate} - z_{false\ alarm\ rate}$$

where the hit and false alarm rates were z-transformed with zero mean and unit variance.

The neuronal d′ was calculated as

$$d'_{neu} = \frac{\mu_{pref} - \mu_{null}}{\sqrt{\frac{\sigma^2_{pref} + \sigma^2_{null}}{2}}}$$

where $\mu_{pref}$ and $\mu_{unlll}$ are the means of the preferred and null direction responses, and $\sigma^2_{pref}$ and $\sigma^2_{null}$ are the variances (*Green and Swets, 1966*). To quantify the neuronal selectivity of both the single neurons and the population, we used the firing rate during the 100200 ms interval after stimulus onset to calculate the d′. This interval was chosen because the firing rates in response to the preferred and null directions were significantly different (*Figure 2—figure supplement 2E*; p<0.05, t-test), and spikes during this time window were significantly correlated with the animals' behavioral choices (*Figure 2—figure supplement 2C*); other time windows between 60–300 ms did not result in differences in the results reported here.

To quantify surround suppression in both psychophysics and neural responses, we first calculated d′ for each stimulus size. The resulting size-tuning curves were fitted with the DoE function (*DeAngelis and Uka, 2003*) (*Figure 2B*):

$$A_e erf\left(\frac{x_c}{s_e}\right) - A_i erf\left(\frac{x_c}{s_e + s_i}\right) + m$$

where $A_e$ and $A_i$ scale the height of the excitatory center and inhibitory surround, respectively, $s_e$ and $s_i$ are the excitatory and inhibitory sizes, and $m$ is the baseline firing rate of the cell, which is set to 0 for the psychophysical and neural selectivity functions.

The suppression index ($SI_{neu}$) for each neuronal size tuning curve was then calculated as $SI_{neu} = (d'_m - d'_L)/d'_m$, where $d'_m$ is the maximum selectivity across responses to different stimulus sizes, and $d'_L$ is the selectivity observed at the largest size. The psychophysical suppression index $SI_{psy}$ was calculated analogously, using psychophysical selectivity rather than neuronal selectivity. Since using the raw responses is sensitive to noise at both the maximum response and the response at the largest size, we used the values from the DoE fits for SI calculations.

Choice probabilities (CP) were used to quantify the relationship between behavioral choice and response variability (*Britten et al., 1996*). For an identical stimulus, the responses can be grouped into two distributions based on whether the monkeys made the choice that corresponds to the neuron's preferred direction, or the null direction (*Figure 2—figure supplement 2A*). As long as the monkeys made at least five choices for each direction, ROC values were calculated from these response distributions, and the area underneath the ROC curve was taken as the CP value (*Figure 2—figure supplement 2B*). The single CP for each neuron was computed by averaging the CP across all stimulus conditions. The alternative method of z-scoring the data for each stimulus conditions and then combining them into a single pair of distributions for preferred and null choices can underestimate the CP when the number of choices for preferred and null directions differs across stimulus conditions (*Kang and Maunsell, 2012*).

## Noise and signal correlations

Noise correlation ($r_{noise}$) was computed as the Pearson correlation coefficient (ranging from -1 to 1) of the trial-by-trial responses of two simultaneously recorded neurons (*Zohary et al., 1994*). For each size and direction combination, responses were z-scored by subtracting the mean response and dividing by the s.d. across stimulus repetitions. This operation removed the effect of size and direction on the mean response, such that $r_{noise}$ measured only correlated trial-to-trial fluctuations around the mean response. To prevent correlations driven by outliers, we only considered trials on which the responses were within ±3 s.d. of the mean (*Zohary et al., 1994*). We also normalized for slow changes in the responses in blocks of 20 trials (*Zohary et al., 1994*).

Signal correlation ($r_{signal}$) was computed as the Pearson correlation coefficient (ranging from -1 to 1) between size tuning curves of preferred and null directions for two simultaneously recorded neurons. Size tuning curves were constructed by plotting mean firing rates as a function of size for preferred and null directions. In addition, we calculated an alternative measure of $r_{signal}$ based on the similarity in direction tuning between the two neurons, and found similar trends between the neuron pairs (*Figure 4—figure supplement 1D*).

As the measure of $r_{noise}$ can depend on the firing rates of the neuron pairs (*Cohen and Kohn, 2011*), we created matched rate distributions of SS-SS and NS-NS pairs by subsampling from the original distributions in *Figure 4A*. We first created distributions of the geometric means of SS-SS and NS-NS pairs and then resampled randomly to create sub-distributions with equal amounts of data in each bin (*Ruff and Cohen, 2014*). We resampled 10,000 times and calculated the slope of the $r_{noise}$ vs. $r_{signal}$ fit of each sub-distribution. The distribution of SS-SS and NS-NS slopes are shown in *Figure 4B*.

## Simulations of population selectivity

The data and Matlab code to generate *Figure 5C,D and E* are available at http://packlab.mcgill.ca/suppression data and code.zip. For all simulations, we considered a population of MT neurons with different receptive field positions and different preferences for stimulus size. The RF locations were determined by fitting a spatial Gaussian to the neuronal response over a 5 x 5 grid. For neurons with RFs within 5° radius of the stimulus center, the responses to different sizes were taken from the size-tuning curves of the actual MT neurons. For neurons with RFs that were not within 5° radius of the stimulus center, we shifted the size-tuning curves by the same proportion as the RF offset, so that a larger stimulus was required to generate the equivalent level of activation. We estimated that the shift in the size-tuning curve is roughly proportional to the shift of the stimulus from the RF center. This was determined by measuring the size-tuning with the stimulus placed at different spatial locations (*Figure 2—figure supplement 1B,C*).

The number of neurons activated by each stimulus was determined using the previously measured cortical magnification in MT, $\mathrm{Magnification\,factor} = 6 * \mathrm{eccentricity}^{-0.9}$ (*Van Essen et al., 1981*;

*Erickson et al., 1989*). This maps visual space in degrees into cortical space in millimeters. The integral of cortical space activation yields the cortical footprint (in square millimeters) as a function of stimulus size. The absolute number of neurons can then be obtained by multiplying the cortical footprint by a factor that indicates the number of neurons per millimeter. We set this factor to 20 neurons/mm$^2$, which yielded a range of pool sizes comparable to those used in other studies (*Shadlen et al., 1996*) (*Figure 1B*). The range of pool sizes is in the regime where population sensitivity is saturated (*Figure 4—figure supplement 1B*). We verified that our results are robust with respect to this parameter re-running the simulations with a value of 40 neurons/mm$^2$; the results were qualitatively similar to those reported here.

## Simulations of population coding

All simulations involved extrapolations from the statistics of our neural recordings. To generate the size tuning curves for the preferred and null directions, $S_i(s,\theta)$, for each simulated neuron, we first used the distributions of DoE parameters from all neurons recorded during the discrimination task to estimate the parameters of a multivariate Gaussian distribution. We then randomly sampled from this distribution to obtain DoE parameters that were subsequently converted to tuning curves. The variance, $V_i(s,\theta)$, for each simulated neuron, was generated by multiplying the $S_i(s,\theta)$ with Fano factors randomly sampled from a Gaussian distribution estimated from the measured Fano factors. For each combination of size and direction in a simulated trial, the response of the $i^{th}$ neuron was generated by randomly drawing a value from a Gaussian distribution having the same mean, $S_i(s,\theta)$, and variance, $V_i(s,\theta)$, as the generated tuning curve

$$R_i(s,\theta) = S_i(s,\theta) + x_i \sqrt{V_i(s,\theta)}$$

where $x$ is a vector of independent random deviates with zero mean and unit variance. This procedure generated a set of responses in which each neuron's noise was independent.

To reproduce the relationship between $r_{noise}$, $r_{signal}$ and surround suppression, the covariance matrix, $r_{noise}$ between neurons $i$ and $j$ was assigned according to

$$r_{noise(i,j)} = SI\ dependency_{(i,j)} \times m \times r_{signal(i,j)} + b$$

where $r_{signal}$ represents the signal correlation between size tuning curves of preferred and null directions for a pair of neurons. The slope $m$ and intercept $b$ were acquired from a linear regression fit to the measured relationship between $r_{noise}$ and $r_{signal}$ for all pairs of neurons. The *SI dependency* term was set to 1 in the no SI modulation condition (*Figure 5C,E*, magenta). In the SI dependency condition (*Figure 5C,E*, cyan) we estimated the dependency empirically from the data as,

$$SI\ dependency_{(i,j)} = 1.3\left( \max_{i,j}\big(SI_i + SI_j\big) - SI_i - SI_j - 2 \right)$$

where $SI_i$ and $SI_j$ were the suppression indices for neurons $i$ and $j$, respectively. When the joint SI of the neuron pairs is high, the value of *SI dependency* will be low, and vice versa, capturing the modulation of the $r_{noise}$ on $r_{signal}$ slope by surround suppression. The constants, 1.3 and 2, were determined using a least squares method to obtain the closest slopes for the 3 groups of neuron pairs in *Figure 4A*. In each iteration of the simulation, we sampled the noise correlation structure from a Wishart distribution with maximum variance around the empirical means.

After assuming the covariance matrix, the response simulation becomes

$$R_i(s,\theta) = S_i(s,\theta) + y_i \sqrt{V_i(s,\theta)}$$

where $y$ represents the product of the matrix square root of the covariance matrix with the vector of independent deviates, $x$ (*Shadlen et al., 1996*; *Cohen and Newsome, 2009*; *Liu et al., 2013*). For each simulation, we generated 1000 trials of responses for each neuron, each size, and each direction.

## Decoding

After generating the responses for 1000 trials for a fixed number of neurons, $n$, at each stimulus size, the 1000 responses in $n$-dimensional space were projected onto the axis that connects the mean

responses. This subsequently generated 1D distributions for the preferred and null direction responses. The 1D distribution of preferred and null direction responses was normalized by their variance and the population d' was then computed while the decoder was blinded to their correlations (*Figure 5B*). This is commonly referred to as a factorial decoder; the readout weights the responses depending on the neuronal sensitivity functions and not on their correlations (*Pitkow et al., 2015*).

In addition to this correlation-blind decoder (*Figure 5*), we also explored the performance of an optimal linear estimator that considers not only the responses of neurons, but also the covariance matrix (*Salinas and Abbott, 1994*). The impact of the dependency between surround suppression and correlation structure is smaller, but still present (*Figure 5—figure supplement 2*).

## Acknowledgements

This work was supported by grants from the Ministère du Développement économique de l'Innovation et de l'Exportation (CCP) and Canadian Institutes of Health Research (MOP-115178 (CCP) and CGSD-121719 (LDL)). We would like to thank Julie Coursol and the staff of the Animal Care Facility (Montreal Neurological Institute) for excellent technical support, and R Born for comments on earlier versions of the manuscript.

## Additional information

### Funding

| Funder | Grant reference number | Author |
| --- | --- | --- |
| Canadian Institutes of Health Research | MOP-115178 | Christopher C Pack |
| Ministere du Developpement economique de l'Innovation et de l'Exportation | | Christopher C Pack |
| Canadian Institutes of Health Research | CGSD-121719 | Liu D Liu |

The funders had no role in study design, data collection and interpretation, or the decision to submit the work for publication.

### Author contributions

LDL, Conception and design, Acquisition of data, Analysis and interpretation of data, Drafting or revising the article; RMH, Analysis and interpretation of data, Drafting or revising the article; CCP, Conception and design, Analysis and interpretation of data, Drafting or revising the article

### Author ORCIDs

Liu D Liu, http://orcid.org/0000-0002-2213-8057

### Ethics

Animal experimentation: All procedures conformed to the regulations established by the Canadian Council on Animal Care and were approved by the Institutional Animal Care Committee of the Montreal Neurological Institute (Protocol number: 5031).

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
