## [Decision Letter]

Thank you for submitting your article "A neural basis for the spatial suppression of visual motion perception" for consideration by *eLife*. Your article has been reviewed by Jochen Ditterich and Adam Morris, and the evaluation has been overseen by Joshua Gold as the Reviewing Editor and David Van Essen as the Senior Editor.

The reviewers have discussed the reviews with one another and the Reviewing Editor has drafted this decision to help you prepare a revised submission.

Summary:

This paper seeks to provide a neural and computational explanation for a perceptual phenomenon in which human observers show reduced perceptual sensitivity for visual motion with larger stimuli. This spatial suppression is surprising because a larger stimulus would be expected to recruit more neurons and therefore provide more information for the perceptual decision. The authors demonstrate that monkeys also show this perceptual phenomenon. They characterize the responses of individual motion-sensitive neurons in area MT and show that, while these neurons demonstrate a wide range of surround suppression strengths, the suppression is, on average, much weaker than the one that is observed behaviorally. This rules out that the behavioral effect could simply be a consequence of an overall suppressed population response with increasing stimulus size. A choice probability analysis is used to demonstrate that neurons with stronger surround suppression do not make a larger contribution to the animals' choices. Using paired recordings, the authors further show that neurons with weak surround suppression, i.e., neurons that would, in principle, be beneficial for processing larger stimuli, are particularly affected by information-limiting noise correlations. As a consequence, the nervous system cannot benefit as much from pooling across these neurons when reading out the sensory information for the purpose of making a decision. Finally, assuming that the monkeys primarily relied on neurons with receptive fields close to the fovea (which is also supported by a choice probability analysis), the authors develop a computational model that is consistent with the behavioral and neural data and supports the idea that the suppression arises from three main factors: specific patterns of correlation among neurons in area MT, limiting the amount of information despite growing population size; surround suppression; and a readout mechanism that gives higher weight to central locations and "ignores" peripheral information.

The two reviewers and the Reviewing Editor all agreed that this is an excellent paper. The motivation for the experiments is well grounded in current theories of population coding; the experimental design is solid and the data quality is high; the modeling is thoughtful and well anchored by their empirical observations; the main conclusions are well supported by their data and are of broad interest to the community; the paper is well written and statistics are applied appropriately.

Essential revisions:

The only major concern raised concerned the presentation of the modeling results. The reviewers felt that this could be improved considerably, to help with the readability of the paper and to help readers better appreciate the relationship between the model and the experimental data. In particular:

1) At least some of the more intuitive descriptions of the model could be moved from the Methods to the Results section (e.g., a shorter version of the information in the subsection “Simulations of population coding”), so those intuitions are available to the reader as the modeling results are being presented.

2) A more direct presentation of psychophysical and simulated results could be useful; e.g. distributions of measured and simulated d-primes and SI's.

3) A more in-depth treatment of the principle shown in Figure 5: how do the model's predictions depend on various model settings? What about parametric variations of the relationship between correlations and surround suppression? How do those ranges compare to measured psychophysical values?

---

## [Author Response]

Essential revisions:

The only major concern raised concerned the presentation of the modeling results. The reviewers felt that this could be improved considerably, to help with the readability of the paper and to help readers better appreciate the relationship between the model and the experimental data. In particular:

1) At least some of the more intuitive descriptions of the model could be moved from the Methods to the Results section (e.g., a shorter version of the information in the subsection “Simulations of population coding”), so those intuitions are available to the reader as the modeling results are being presented.

Thanks for this suggestion. We now include a short description of the model at the beginning of the Results section:

“A detailed description of the modeling approach is given in the Methods. In brief, we generated populations of synthetic neurons by sampling neural properties from a joint truncated Normal distribution over tuning curve parameters inferred from our measurements. […] The predicted model suppression is the key metric that we are interested in, and its dependency on the key model parameters are explored in Figure 5—figure supplement 4 and Figure 5.”

2) A more direct presentation of psychophysical and simulated results could be useful; e.g. distributions of measured and simulated d-primes and SI's.

We now include the distributions of measured and simulated d-primes and SIs in Figure 5—figure supplement 3. We note that the simulations consistently outperform the monkeys. By incorporating pooling noise into our simulations, we could scale behavioral performance (d’) down to match experimental data. However, this would result in the same SIs for the simulations. Since the main purpose of the simulations is to match the distribution of the SIs to the psychophysical measurements, we avoided introducing this additional factor into our simulations.

3) A more in-depth treatment of the principle shown in Figure 5: how do the model's predictions depend on various model settings? What about parametric variations of the relationship between correlations and surround suppression? How do those ranges compare to measured psychophysical values?

We have added a more in-depth treatment of the other main model settings in Figure 5—figure supplement 4. As the reviewers noted, the suppression shown by the model is due to three factors: 1) surround suppression in single neurons; 2) the relationship between correlations and surround suppression; and 3) a readout that gives higher weight to central locations. We only included the parametric variation of the third factor in Figure 5.

In the revision, we have further explored the influence of surround suppression in single neurons and the relationship between correlations and surround suppression. To do this, we varied the degree of surround suppression in single neurons by simply manipulating the amplitude of the inhibitory error function from the preferred direction responses across all sizes. Unsurprisingly, the suppression of the model positively correlated with the surround suppression in the single neurons (Figure 5—figure supplement 4).

The relationship between correlations and surround suppression can then be summarized by the following term in the paper.

SI dependency(i,j)=1.3(maxi,j(SIi+SIj)−SIi−SIj−2)The parameters are the slope and offset, which are set to their measured values (1.3 ± 0.3 and 2 ± 0.2, 95% confidence interval from the curve fit) for the main simulations. The consequences of varying them are shown in Figure 5—figure supplement 4. Larger slopes lead to larger simulated SIs, for reasons that are detailed in the paper. There is no obvious influence of the offset parameter.